# An Ensemble Diversity Approach to Supervised Binary Hashing

**Miguel Á. Carreira-Perpiñán**
EECS, University of California, Merced
mcarreira-perpinan@ucmerced.edu

**Ramin Raziperchikolaei**
EECS, University of California, Merced
rraziperchikolaei@ucmerced.edu

## Abstract

Binary hashing is a well-known approach for fast approximate nearest-neighbor search in information retrieval. Much work has focused on affinity-based objective functions involving the hash functions or binary codes. These objective functions encode neighborhood information between data points and are often inspired by manifold learning algorithms. They ensure that the hash functions differ from each other through constraints or penalty terms that encourage codes to be orthogonal or dissimilar across bits, but this couples the binary variables and complicates the already difficult optimization. We propose a much simpler approach: we train each hash function (or bit) independently from each other, but introduce diversity among them using techniques from classifier ensembles. Surprisingly, we find that not only is this faster and trivially parallelizable, but it also improves over the more complex, coupled objective function, and achieves state-of-the-art precision and recall in experiments with image retrieval.

Information retrieval tasks such as searching for a query image or document in a database are essentially a nearest-neighbor search [33]. When the dimensionality of the query and the size of the database is large, approximate search is necessary. We focus on binary hashing [17], where the query and database are mapped onto low-dimensional binary vectors, where the search is performed. This has two speedups: computing Hamming distances (with hardware support) is much faster than computing distances between high-dimensional floating-point vectors; and the entire database becomes much smaller, so it may reside in fast memory rather than disk (for example, a database of 1 billion real vectors of dimension 500 takes 2 TB in floating point but 8 GB as 64-bit codes).

Constructing hash functions that do well in retrieval measures such as precision and recall is usually done by optimizing an affinity-based objective function that relates Hamming distances to supervised neighborhood information in a training set. Many such objective functions have the form of a sum of pairwise terms that indicate whether the training points $\mathbf{x}_n$ and $\mathbf{x}_m$ are neighbors:

$$\min_{\mathbf{h}} \mathcal{L}(\mathbf{h}) = \sum_{n,m=1}^{N} L(\mathbf{z}_n, \mathbf{z}_m; y_{nm}) \text{ where } \mathbf{z}_m = \mathbf{h}(\mathbf{x}_m), \ \mathbf{z}_n = \mathbf{h}(\mathbf{x}_n).$$

Here, $\mathbf{X} = (\mathbf{x}_1, \ldots, \mathbf{x}_N)$ is the dataset of high-dimensional feature vectors (e.g., SIFT features of an image), $\mathbf{h} \colon \mathbb{R}^D \to \{-1, +1\}^b$ are $b$ binary hash functions and $\mathbf{z} = \mathbf{h}(\mathbf{x})$ is the $b$-bit code vector for input $\mathbf{x} \in \mathbb{R}^D$, $\min_{\mathbf{h}}$ means minimizing over the parameters of the hash function $\mathbf{h}$ (e.g. over the weights of a linear SVM), and $L(\cdot)$ is a loss function that compares the codes for two images (often through their Hamming distance $\|\mathbf{z}_n - \mathbf{z}_m\|$) with the ground-truth value $y_{nm}$ that measures the affinity in the original space between the two images $\mathbf{x}_n$ and $\mathbf{x}_m$ (distance, similarity or other measure of neighborhood). The sum is often restricted to a subset of image pairs $(n, m)$ (for example, within the $k$ nearest neighbors of each other in the original space), to keep the runtime low. The output of the algorithm is the hash function $\mathbf{h}$ and the binary codes $\mathbf{Z} = (\mathbf{z}_1, \ldots, \mathbf{z}_N)$ for the training points, where $\mathbf{z}_n = \mathbf{h}(\mathbf{x}_n)$ for $n = 1, \ldots, N$. Examples of these objective functions are Supervised Hashing with Kernels (KSH) [28], Binary Reconstructive Embeddings (BRE) [21] and the binary Laplacian loss (an extension of the Laplacian Eigenmaps objective; [2]) where $L(\mathbf{z}_n, \mathbf{z}_m; y_{nm})$ is:

$$\text{KSH: } (\mathbf{z}_n^T \mathbf{z}_m - b y_{nm})^2 \qquad \text{BRE: } \left(\tfrac{1}{b} \|\mathbf{z}_n - \mathbf{z}_m\|^2 - y_{nm}\right)^2 \qquad \text{LAP: } y_{nm} \|\mathbf{z}_n - \mathbf{z}_m\|^2 \qquad (1)$$

where for KSH $y_{nm}$ is 1 if $\mathbf{x}_n$, $\mathbf{x}_m$ are similar and $-1$ if they are dissimilar; for BRE $y_{nm} = \tfrac{1}{2} \|\mathbf{x}_n - \mathbf{x}_m\|^2$ (where the dataset $\mathbf{X}$ is normalized so the Euclidean distances are in $[0, 1]$); and for the Laplacian loss $y_{nm} > 0$ if $\mathbf{x}_n$, $\mathbf{x}_m$ are similar and $< 0$ if they are dissimilar ("positive" and "negative" neighbors). Other examples of these objectives include models developed for dimension reduction, be they spectral such as Locally Linear Embedding [32] or Anchor Graphs [27], or non-linear such as the Elastic Embedding [7] or $t$-SNE; as well as objectives designed specifically for binary hashing, such as Semi-supervised sequential Projection Learning Hashing (SPLH) [34]. They all can produce good hash functions. We will focus on the Laplacian loss in this paper.

In designing these objective functions, one needs to eliminate two types of trivial solutions. 1) In the Laplacian loss, mapping all points to the same code, i.e., $\mathbf{z}_1 = \cdots = \mathbf{z}_N$, is the global optimum of the positive neighbors term (this also arises if the codes $\mathbf{z}_n$ are real-valued, as in Laplacian eigenmaps). This can be avoided by having negative neighbors. 2) Having all hash functions (all $b$ bits of each vector) being identical to each other, i.e., $z_{n1} = \cdots = z_{nb}$ for each $n = 1, \ldots, N$. This can be avoided by introducing constraints, penalty terms or other mathematical devices that couple the $b$ bits. For example, in the Laplacian loss (1) we can encourage codes to be orthogonal through a constraint $\mathbf{Z}^T \mathbf{Z} = N \mathbf{I}$ [35] or a penalty term $\|\mathbf{Z}^T \mathbf{Z} - N \mathbf{I}\|^2$ (with a hyperparameter that controls the weight of the penalty) [14], although this generates dense matrices of $N \times N$. In the KSH or BRE (1), squaring the dot product or Hamming distance between the codes couples the $b$ bits.

An important downside of these approaches is the difficulty of their optimization. This is due to the fact that the objective function is nonsmooth (implicitly discrete) because of the binary output of the hash function. There is a large number of such binary variables ($bN$), a larger number of pairwise interactions ($\mathcal{O}(N^2)$, less if using sparse neighborhoods) and the variables are coupled by the said constraints or penalty terms. The optimization is approximated in different ways. Most papers ignore the binary nature of the $\mathbf{Z}$ codes and optimize over them as real values, then binarize them by truncation (possibly with an optimal rotation; [16]), and finally fit a classifier (e.g. linear SVM) to each of the $b$ bits separately. For example, for the Laplacian loss with constraints this involves solving an eigenproblem on $\mathbf{Z}$ as in Laplacian eigenmaps [2, 35, 36], or approximated using landmarks [27]. This is fast, but relaxing the codes in the optimization is generally far from optimal. Some recent papers try to respect the binary nature of the codes during their optimization, using techniques such as alternating optimization, min-cut and GraphCut [4, 14, 26] or others [25], and then fit the classifiers, or use alternating optimization directly on the hash function parameters [28]. Even more recently, one can optimize jointly over the binary codes and hash functions [8, 14, 31]. Most of these approaches are slow and limited to small datasets (a few thousand points) because of the quadratic number of pairwise terms in the objective.

We propose a different, much simpler approach. Rather than coupling the $b$ hash functions into a single objective function, we train each hash function *independently from each other and using a single-bit objective function of the same form*. We show that we can avoid trivial solutions by injecting diversity into each hash function's training using techniques inspired from classifier ensemble learning. Section 1 discusses relevant ideas from the ensemble learning literature, section 2 describes our *independent Laplacian hashing* algorithm, section 3 gives evidence with image retrieval datasets that this simple approach indeed works very well, and section 4 further discusses the connection between hashing and ensembles.

## 1 Ideas from learning classifier ensembles

At first sight, optimizing Laplacian loss without constraints does not seem like a good idea: since $\|\mathbf{z}_n - \mathbf{z}_m\|^2$ separates over the $b$ bits, we obtain $b$ independent identical objectives, one over each hash function, and so they all have the same global optimum. And, if all hash functions are equal, they are equivalent to using just one of them, which will give a much lower precision/recall. In fact, the very same issue arises when training an ensemble of classifiers [10, 22]. Here, we have a training set of input vectors and output class labels, and want to train several classifiers whose outputs are then combined (usually by majority vote). If the classifiers are all equal, we gain nothing over a single classifier. Hence, it is necessary to introduce *diversity* among the classifiers so that they disagree in their predictions. The ensemble learning literature has identified several mechanisms to inject diversity. The most important ones that apply to our binary hashing setting are as follows:

**Using different data for each classifier** This can be done by: 1) Using different feature subsets for each classifier. This works best if the features are somewhat redundant. 2) Using different

training sets for each classifier. This works best for unstable algorithms (whose resulting classifier is sensitive to small changes in the training data), such as decision trees or neural nets, and unlike linear or nearest neighbor classifiers. A prominent example is bagging [6], which generates bootstrap datasets and trains a model on each.

**Injecting randomness in the training algorithm** This is only possible if local optima exist (as for neural nets) or if the algorithm is randomized (as for decision trees). This can be done by using different initializations, adding noise to the updates or using different choices in the randomized operations (e.g. the choice of split in decision trees, as in random forests; [5]).

**Using different classifier models** For example, different parameters (e.g. the number of neighbors in a nearest-neighbor classifier), different architectures (e.g. neural nets with different number of layers or hidden units), or different types of classifiers altogether.

## 2   Independent Laplacian Hashing (ILH) with diversity

The connection of binary hashing with ensemble learning offers many possible options, in terms of the choice of type of hash function ("base learner"), binary hashing (single-bit) objective function, optimization algorithm, and diversity mechanism. In this paper we focus on the following choices. We use linear and kernel SVMs as hash functions. Without loss of generality (see later), we use the Laplacian objective (1), which for a single bit takes the form

$$E(\mathbf{z}) = \sum_{n,m=1}^{N} y_{nm}(z_n - z_m)^2, z_n = h(\mathbf{x}_n) \in \{-1, 1\}, n = 1, \dots, N. \tag{2}$$

To optimize it, we use a two-step approach, where we first optimize (2) over the $N$ bits and then learn the hash function by fitting to it a binary classifier. (It is also possible to optimize over the hash function directly with the method of auxiliary coordinates; [8, 31], which essentially iterates over optimizing (2) and fitting the classifier.) The Laplacian objective (2) is NP-complete if we have negative neighbors (i.e., some $y_{nm} < 0$). We approximately optimize it using a min-cut algorithm (as implemented in [4]) applied in alternating fashion to submodular blocks as described in Lin et al. [24]. This first partitions the $N$ points into disjoint groups containing only nonnegative weights. Each group defines a submodular function (specifically, quadratic with nonpositive coefficients) whose global minimum can be found in polynomial time using min-cut. The order in which the groups are optimized over is randomized at each iteration (this improves over using a fixed order). The approximate optimizer found depends on the initial $\mathbf{z} \in \{-1, 1\}^N$.

Finally, we consider three types of *diversity mechanism* (as well as their combination):

**Different initializations (ILHi)** Each hash function is initialized from a random $N$-bit vector $\mathbf{z}$.

**Different training sets (ILHt)** Each hash function uses a training set of $N$ points that is different and (if possible) disjoint from that of other hash functions. We can afford to do this because in binary hashing the training sets are potentially very large, and the computational cost of the optimization limits the training sets to a few thousand points. Later we show this outperforms using bootstrapped training sets.

**Different feature subsets (ILHf)** Each hash function is trained on a random subset of $1 \leq d \leq D$ features sampled without replacement (so the $d$ features are distinct). The subsets corresponding to different hash functions may overlap.

These mechanisms are applicable to other objective functions beyond (2). We could also use the same training set but construct differently the weight matrix in (2) (e.g. using different numbers of positive and negative neighbors).

**Equivalence of objective functions in the single-bit case** Several binary hashing objectives that differ in the general case of $b > 1$ bits become essentially identical in the $b = 1$ case. For example, expanding the pairwise terms in (1) (noting that $z_n^2 = 1$ if $z_n \in \{-1, +1\}$) gives $L(z_n, z_m; y_{nm})$ as

KSH: $-2y_{nm}z_n z_m$+constant   BRE: $-4(2-y_{nm})z_n z_m$+constant   LAP: $-2y_{nm}z_n z_m$+constant.

So all the three objectives are in fact identical and can be written in the form of a binary quadratic function without linear term (or a Markov random field with quadratic potentials only):

$$\min_{\mathbf{z}} E(\mathbf{z}) = \mathbf{z}^T \mathbf{A} \mathbf{z} \quad \text{with} \quad \mathbf{z} \in \{-1, +1\}^N \tag{3}$$

with an appropriate, data-dependent neighborhood symmetric matrix $\mathbf{A}$ of $N \times N$. This problem is NP-complete in general [3, 13, 18], when $\mathbf{A}$ has both positive and negative elements, as well as zeros. It is submodular if $\mathbf{A}$ has only nonpositive elements, in which case it is equivalent to a min-cut/max-flow problem and it can be solved in polynomial time [3].

More generally, any function of a binary vector $\mathbf{z}$ that has the form $E(\mathbf{z}) = \sum_{n,m=1}^{N} f_{nm}(z_n, z_m)$ and which only depends on Hamming distances between bits $z_n, z_m$ can be written as $f_{nm}(z_n, z_m) = a_{nm} z_n z_m + b_{nm}$. Even more, an arbitrary function of 3 binary variables that depends only on their Hamming distances can be written as a quadratic function of the 3 variables. However, for 4 variables or more this is not generally true (see supplementary material).

**Computational advantages**   Training the hash functions independently has some important advantages. First, training the $b$ functions can be parallelized perfectly. This is a speedup of one to two orders of magnitude for typical values of $b$ (32 to 200 in our experiments). Coupled objective functions such as KSH do not exhibit obvious parallelism, because they are trained with alternating optimization, which is inherently sequential.

Second, even in a single processor, $b$ binary optimizations over $N$ variables each is generally easier than one binary optimization over $bN$ variables. This is so because the search spaces contain $b2^N$ and $2^{bN}$ states, respectively, so enumeration is much faster in the independent case (even though it is still impractical). If using an approximate polynomial-time algorithm, the independent case is also faster if the runtime is superlinear on the number of variables: the asymptotic runtimes will be $\mathcal{O}(bN^\alpha)$ and $\mathcal{O}((bN)^\alpha)$ with $\alpha > 1$, respectively. This is the case for the best practical GraphCut [4] and max-flow/min-cut algorithms [9].

Third, the solution exhibits "nesting", that is, to get the solution for $b+1$ bits we just need to take a solution with $b$ bits and add one more bit (as happens with PCA). This is unlike most methods based on a coupled objective function (such as KSH), where the solution for $b+1$ bits cannot be obtained by adding one more bit, we have to solve for $b+1$ bits from scratch.

For ILHf, both the training and test time are lower than if using all $D$ features for each hash function. The test runtime for a query is $d/D$ times smaller.

**Model selection for the number of bits** $b$   Selecting the number of bits (hash functions) to use has not received much attention in the binary hashing literature. The most obvious way to do this would be to maximize the precision on a test set over $b$ (cross-validation) subject to $b$ not exceeding a preset limit (so applying the hash function is fast with test queries). The nesting property of ILH makes this computationally easy: we simply keep adding bits until the test precision stabilizes or decreases, or until we reach the maximum $b$. We can still benefit from parallel processing: if $P$ processors are available, we train $P$ hash functions in parallel and evaluate their precision, also in parallel. If we still need to increase $b$, we train $P$ more hash functions, etc.

# 3   Experiments

We use the following labeled datasets (all using the Euclidean distance in feature space): (1) CIFAR [19] contains $60\,000$ images in 10 classes. We use $D = 320$ GIST features [30] from each image. We use $58\,000$ images for training and $2\,000$ for test. (2) Infinite MNIST [29]. We generated, using elastic deformations of the original MNIST handwritten digit dataset, $1\,000\,000$ images for training and $2\,000$ for test, in 10 classes. We represent each image by a $D = 784$ vector of raw pixels. The supplementary material contains experiments on additional datasets.

Because of the computational cost of affinity-based methods, previous work has used training sets limited to a few thousand points [14, 21, 25, 28]. Unless otherwise indicated, we train the hash functions in a subset of $5\,000$ points of the training set, and report precision and recall by searching for a test query on the entire dataset (the base set). As hash functions (for each bit), we use linear SVMs (trained with LIBLINEAR; [12]) and kernel SVMs (with $500$ basis functions centered at a random subset of training points). We report precision and recall for the test set queries using as ground truth (set of true neighbors in original space) all the training points with the same label as the query. The retrieved set contains the $k$ nearest neighbors of the query point in the Hamming space. We report precision for different values of $k$ to test the robustness of different algorithms.

**Diversity mechanisms with ILH**   To understand the effect of diversity, we evaluate the 3 mechanisms ILHi, ILHt and ILHf, and their combination ILHitf, over a range of number of bits $b$ (32 to 128) and training set size $N$ ($2\,000$ to $20\,000$). As baseline coupled objective, we use KSH [28] but using the same two-step training as ILH: first we find the codes using the alternating min-cut method described earlier (initialized from an all-ones code, and running one iteration of alternating min-cut) and then we fit the classifiers. This is faster and generally finds better optima than the original KSH optimization [26]. We denote it as KSHcut.

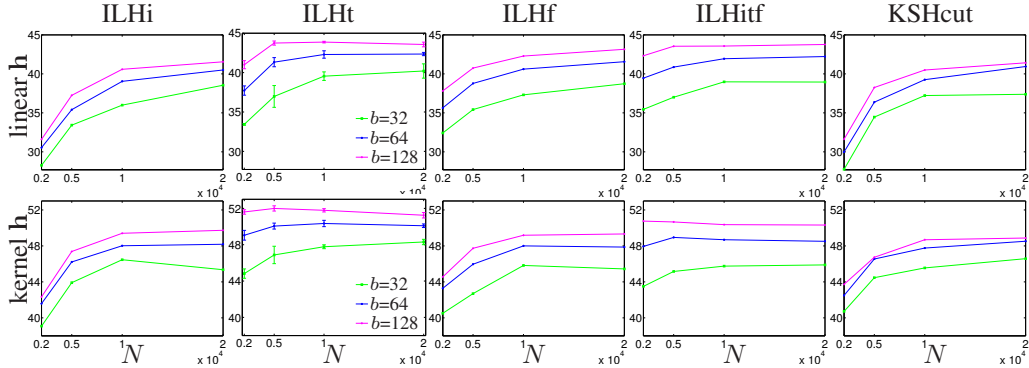

Figure 1: Diversity mechanisms vs baseline (KSHcut). Precision on CIFAR dataset, as a function of the training set size $N$ (2,000 to 20 000) and number of bits $b$ (32 to 128). Ground truth: all points with the same label as the query. Retrieved set: $k = 500$ nearest neighbors of the query. Errorbars shown only for ILHt (over 5 random training sets) to avoid clutter. *Top to bottom*: linear and kernel hash functions. *Left to right*: diversity mechanisms, their combination, and the baseline KSHcut.

Fig. 1 shows the results. *The clearly best diversity mechanism is ILHt*, which works better than the other mechanisms, even when combined with them, and significantly better than KSHcut. We explain this as follows. Although all 3 mechanisms introduce diversity, ILHt has a distinct advantage (also over KSHcut): it effectively uses $b$ times as much training data, because each hash function has its own disjoint dataset. Using $bN$ training points in KSHcut would be orders of magnitude slower. ILHt is equal or even better than the combined ILHitf because 1) since there is already enough diversity in ILHt, the extra diversity from ILHi and ILHf does not help; 2) ILHf uses less data (it discards features), which can hurt the precision; this is also seen in fig. 2 (panel 2). The precision of all methods saturates as $N$ increases; with $b = 128$ bits, ILHt achieves nearly maximum precision with only 5 000 points. In fact, if we continued to increase the per-bit training set size $N$ in ILHt, eventually all bits would use the same training set (containing all available data), diversity would disappear and the precision would drop drastically to the precision of using a single bit ($\approx 12\%$). Practical image retrieval datasets are so large that this is unlikely to occur unless $N$ is very large (which would make the optimization too slow anyway).

Linear SVMs are very stable classifiers known to benefit less from ensembles than less stable classifiers such as decision trees or neural nets [22]. Remarkably, they strongly benefit from the ensemble in our case. This is because each hash function is solving a different classification problem (different output labels), so the resulting SVMs are in fact quite different from each other. The conclusions for kernel hash functions are similar. In fig. 1, the kernel functions are using the same, common 500 centers for the radial basis functions. Nonlinear classifiers are less stable than linear ones. In our case they do not benefit much more than linear SVMs from the diversity. They do achieve higher precision since they are more powerful models. See supplementary material for more results.

Fig. 2 shows the results on infinite MNIST dataset (see supp. mat for the results on CIFAR). Panel 1 shows the results in ILHf of varying the number of features $1 \leq d \leq D$ used by each hash function. Intuitively, very low $d$ is bad because each classifier receives too little information and will make near-random codes. Indeed, for low $d$ the precision is comparable to that of LSH (random projections) in panel 4. Very high $d$ will also work badly because it would eliminate the diversity and drop to the precision of a single bit for $d = D$. This does not happen because there is an additional source of diversity: the randomization in the alternating min-cut iterations. This has an effect similar to that of ILHi, and indeed a comparable precision. The highest precision is achieved with a proportion $d/D \approx 30\%$ for ILHf, indicating some redundancy in the features. When combined with the other diversity mechanisms (ILHitf, panel 2), the highest precision occurs for $d = D$, because diversity is already provided by the other mechanisms, and using more data is better.

Fig. 2 (panel 3) shows the results of constructing the $b$ training sets for ILHt as a random sample from the base set such that they are "bootstrapped" (sampled with replacement), "disjoint" (sampled without replacement) or "random" (sampled without replacement but reset for each bit, so the training sets may overlap). As expected, "disjoint" (closely followed by "random") is consistently and notably better than "bootstrap" because it introduces more independence between the hash functions and learns from more data overall (since each hash function uses the same training set size).

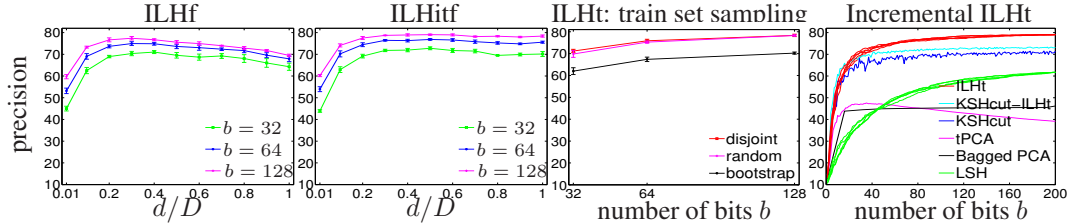

Figure 2: *Panels 1–2*: effect of the proportion of features $d/D$ used in ILHf and ILHitf. *Panel 3*: bootstrap vs random vs disjoint training sets in ILHt. *Panel 4*: precision as a function of the number of hash functions $b$ for different methods. All results show precision using a training set of $N = 5\,000$ points of infinite MNIST dataset. Errorbars over 5 random training sets. Ground truth: all points with the same label as the query. Retrieved set: $k = 10\,000$ nearest neighbors of the query.

**Precision as a function of** $b$ Fig. 2 (panel 4) shows the precision (in the test set) as a function of the number of bits $b$ for ILHt, where the solution for $b + 1$ bits is obtained by adding a new bit to the solution for $b$. Since the hash functions obtained depend on the order in which we add the bits, we show 5 such orders (red curves). Remarkably, *the precision increases nearly monotonically* and continues increasing beyond $b = 200$ bits (note the prediction error in bagging ensembles typically levels off after around 25–50 decision trees; [22, p. 186]). This is (at least partly) because the effective training set size is proportional to $b$. The variance in the precision decreases as $b$ increases. In contrast, for KSHcut the variance is larger and the precision barely increases after $b = 80$. The higher variance for KSHcut is due to the fact that each $b$ value involves training from scratch and we can converge to a relatively different local optimum. As with ILHt, adding LSH random projections (again 5 curves for different orders) increases precision monotonically, but can only reach a low precision at best, since it lacks supervision. We also show the curve for thresholded PCA (tPCA), whose precision tops at around $b = 30$ and decreases thereafter. A likely explanation is that high-order principal components essentially capture noise rather than signal, i.e., random variation in the data, and this produces random codes for those bits, which destroy neighborhood information. Bagging tPCA (here, using ensembles where each member has 16 principal components, i.e., 16 bits) [23] does make tPCA improve monotonically with $b$, but the result is still far from competitive. The reason is the low diversity among the ensemble members, because the top principal components can be accurately estimated even from small samples.

Is the precision gap between KSH and ILHt due to an incomplete optimization of the KSH objective, or to bad local optima? We verified that 1) random perturbations of the KSHcut optimum lower the precision; 2) optimizing KSHcut using the ILHt codes as initialization ("KSHcut-ILHt" curve) increases the precision but it still remains far from that of ILHt. This confirms that the optimization algorithm is doing its job, and that *the ILHt diversity mechanism is superior to coupling the hash functions in a joint objective*.

**Are the codes orthogonal?** The result of learning binary hashing is $b$ functions, represented by a matrix $\mathbf{W}_{b \times D}$ of real weights for linear SVMs, and a matrix $\mathbf{Z}_{N \times b}$ of binary $(-1, +1)$ codes for the entire dataset. We define a *measure of code orthogonality* as follows. Define $b \times b$ matrices $\mathbf{C_Z} = \frac{1}{N}\mathbf{Z}^T\mathbf{Z}$ for the codes and $\mathbf{C_W} = \mathbf{W}\mathbf{W}^T$ for the weights (assuming normalized SVM weights). Each $\mathbf{C}$ matrix has entries in $[-1, 1]$, equal to a normalized dot product of codes or weight vectors, and diagonal entries equal to 1. (Note that any matrix $\mathbf{SCS}$ where $\mathbf{S}$ is diagonal with $\pm 1$ entries is equivalent, since reverting a hash function's output does not alter the Hamming distances.) Perfect orthogonality happens when $\mathbf{C} = \mathbf{I}$, and is encouraged by many binary hashing methods.

Fig. 3 shows this for ILHt in CIFAR ($N = 58\,000$ training points of dim. $D = 320$). It plots $\mathbf{C_Z}$ as an image, as well as the histogram of the entries of $\mathbf{C_Z}$ and $\mathbf{C_W}$. The histograms also contain, as a control, the histogram corresponding to normalized dot products of random vectors (of dimension $N$ or $D$, respectively), which is known to tend to a delta function at 0 as the dimension grows. Although $\mathbf{C_W}$ has some tendency to orthogonality as the number of bits $b$ increases, it is clear that, for both codes and weight vectors, the distribution of dot products is wide, far from strict orthogonality. Hence, *enforcing orthogonality does not seem necessary to achieve good hash functions and codes*.

**Comparison with other binary hashing methods** We compare with both the original KSH [28] and its min-cut optimization KSHcut [26], and a representative subset of affinity-based and unsupervised hashing methods: Supervised Binary Reconstructive Embeddings (BRE) [21], Supervised Self-Taught Hashing (STH) [36], Spectral Hashing (SH) [35], Iterative Quantization (ITQ) [16], Bi-

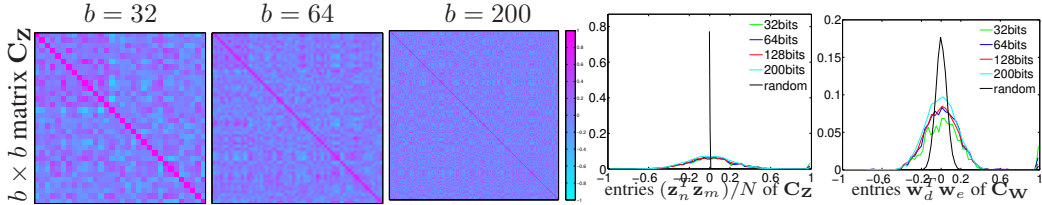

Figure 3: Orthogonality of codes ($b \times b$ images and left histogram) and of hash function weight vectors (right histogram) in CIFAR.

nary Autoencoder (BA) [8], thresholded PCA (tPCA), and Locality-Sensitive Hashing (LSH) [1]. We create affinities $y_{nm}$ for all the affinity-based methods using the dataset labels. For each training point $\mathbf{x}_n$, we use as similar neighbors 100 points with the same labels as $\mathbf{x}_n$; and as dissimilar neighbors 100 points chosen randomly among the points whose labels are different from that of $\mathbf{x}_n$. For all datasets, all the methods are trained using a subset of 5 000 points. Given that KSHcut already performs well [26] and that ILHt consistently outperforms it both in precision and runtime, we expect ILHt to be competitive with the state-of-the-art. Fig. 4 shows this is generally the case, particularly as the number of bits $b$ increases, when ILHt beats all other methods, which are not able to increase precision as much as ILHt does.

**Runtime**   Training a single ILHt hash function (in a single processor) for CIFAR dataset with $N = 2\,000, 5\,000$ and $20\,000$ takes $1.2, 2.8$ and $22.5$ seconds, respectively. This is much faster than other affinity-based hashing methods (for example, for 128 bits with 5 000 points, BRE did not converge after 12 hours). KSHcut is among the faster methods. Its runtime per min-cut pass over a single bit is comparable to ours, but it needs $b$ sequential passes to complete just one alternating optimization iteration, while our $b$ functions can be trained in parallel.

**Summary**   ILHt achieves a remarkably high precision compared to a coupled KSH objective using the same optimization algorithm but introducing diversity by feeding different data to independent hash functions rather than by jointly optimizing over them. It also compares well with state-of-the-art methods in precision/recall, being competitive if few bits are used and the clear winner as more bits are used, and is very fast and embarrassingly parallel.

## 4   Discussion

We have revealed for the first time a connection between supervised binary hashing and ensemble learning that could open the door to many new hashing algorithms. Although we have focused on a specific objective and identified as particularly successful with it a specific diversity mechanism (disjoint training sets), other choices may be better depending on the application. The core idea we propose is the independent training of the hash functions via the introduction of diversity by means other than coupling terms in the objective or constraints. This may come as a surprise in the area of learning binary hashing, where most work has focused on proposing complex objective functions that couple all $b$ hash functions and developing sophisticated optimization algorithms for them.

Another surprise is that orthogonality of the codes or hash functions seems unnecessary. ILHt creates codes and hash functions that do differ from each other but are far from being orthogonal, yet they achieve good precision that keeps growing as we add bits. Thus, introducing diversity through different training data seems a better mechanism to make hash functions differ than coupling the codes through an orthogonality constraint or otherwise. It is also far simpler and faster to train independent single-bit hash functions.

A final surprise is that the wide variety of affinity-based objective functions in the $b$-bit case reduces to a binary quadratic problem in the 1-bit case regardless of the form of the $b$-bit objective (as long as it depends on Hamming distances only). In this sense, there is a unique objective in the 1-bit case.

There has been a prior attempt to use bagging (bootstrapped samples) with truncated PCA [23]. Our experiments show that, while this improves truncated PCA, it performs poorly in supervised hashing. This is because PCA is unsupervised and does not use the user-provided similarity information, which may disagree with Euclidean distances in image space; and because estimating principal components from samples has low diversity. Also, PCA is computationally simple and there is little gain by bagging it, unlike the far more difficult optimization of supervised binary hashing.

Some supervised binary hashing work [28, 34] has proposed to learn the $b$ hash functions sequentially, where the $i$th function has an orthogonality-like constraint to force it to differ from the previ-

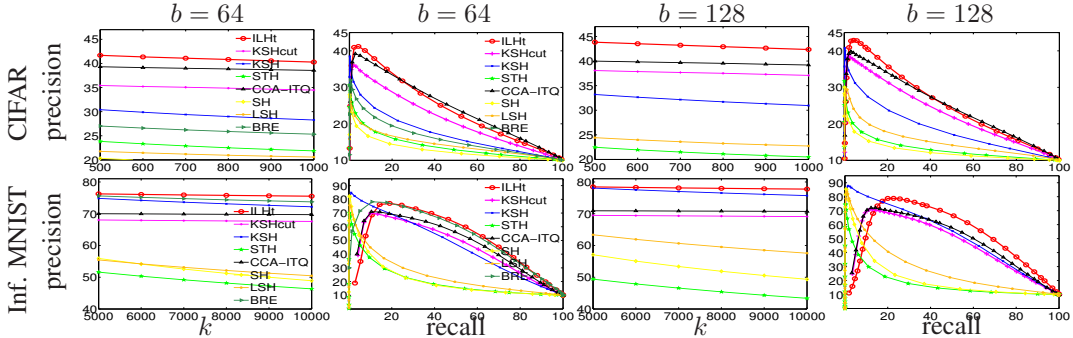

Figure 4: Comparison with binary hashing methods in precision and precision/recall, using linear SVMs as hash functions and different numbers of bits $b$, for CIFAR and Inf. MNIST.

ous functions. Hence, this does not learn the functions independently and can be seen as a greedy optimization of a joint objective over all $b$ functions.

Binary hashing does differ from ensemble learning in one important point: the predictions of the $b$ classifiers ($= b$ hash functions) are not combined into a single prediction, but are instead concatenated into a binary vector (which can take $2^b$ possible values). The "labels" (the binary codes) for the "classifiers" (the hash functions) are unknown, and are implicitly or explicitly learned together with the hash functions themselves. This means that well-known error decompositions such as the error-ambiguity decomposition [20] and the bias-variance decomposition [15] do not apply. Also, the real goal of binary hashing is to do well in information retrieval measures such as precision and recall, but hash functions do not directly optimize this. A theoretical understanding of why diversity helps in learning binary hashing is an important topic of future work.

In this respect, there is also a relation with error-correcting output codes (ECOC) [11], an approach for multiclass classification. In ECOC, we represent each of the $K$ classes with a $b$-bit binary vector, ensuring that $b$ is large enough for the vectors to be sufficiently separated in Hamming distance. Each bit corresponds to partitioning the $K$ classes into two groups. We then train $b$ binary classifiers, such as decision trees. Given a test pattern, we output as class label the one closest in Hamming distance to the $b$-bit output of the $b$ classifiers. The redundant error-correcting codes allow for small errors in the individual classifiers and can improve performance. An ECOC can also be seen as an ensemble of classifiers where we manipulate the output targets (rather than the input features or training set) to obtain each classifier, and we apply majority vote on the final result (if the test output in classifier $i$ is 1, then all classes associated with 1 get a vote). The main benefit of ECOC seems to be in variance reduction, as in other ensemble methods. Binary hashing can be seen as an ECOC with $N$ classes, one per training point, with the ECOC prediction for a test pattern (query) being the nearest-neighbor class codes in Hamming distance. However, unlike in ECOC, in binary hashing the codes are learned so they preserve neighborhood relations between training points. Also, while ideally all $N$ codes should be different (since a collision makes two originally different patterns indistinguishable, which will degrade some searches), this is not guaranteed in binary hashing.

## 5 Conclusion

Much work in supervised binary hashing has focused on designing sophisticated objectives of the hash functions that force them to compete with each other while trying to preserve neighborhood information. We have shown, surprisingly, that training hash functions independently is not just simpler, faster and parallel, but also can achieve better retrieval quality, as long as diversity is introduced into each hash function's objective function. This establishes a connection with ensemble learning and allows one to borrow techniques from it. We showed that having each hash function optimize a Laplacian objective on a disjoint subset of the data works well, and facilitates selecting the number of bits to use. Although our evidence is mostly empirical, the intuition behind it is sound and in agreement with the many results (also mostly empirical) showing the power of ensemble classifiers. The ensemble learning perspective suggests many ideas for future work, such as pruning a large ensemble or using other diversity techniques. It may also be possible to characterize theoretically the performance in precision of binary hashing depending on the diversity of the hash functions.

**Acknowledgments**
Work supported by NSF award IIS–1423515.

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
