[Supplementary Material · nips16a-supp.pdf]

# Supplementary material for:
# An Ensemble Diversity Approach to Supervised Binary Hashing

Miguel Á. Carreira-Perpiñán       Ramin Raziperchikolaei

Electrical Engineering and Computer Science, University of California, Merced

http://eecs.ucmerced.edu

May 20, 2016

### Abstract

This is supplementary material for the "main paper" [3]. We provide the following. 1) A proof for the statements in the main paper (section 2, p. 3) regarding the equivalence of objective functions in the single-bit case. 2) A proof for the statements in the main paper (section 3, p. 6) regarding the orthogonality measure for binary code vectors or hash function weight vectors. 3) Extended experiments (figures 1–4) with additional hash functions, datasets and numbers of bits $b$, and comparing with additional binary hashing methods.

## 1   Equivalence of objective functions in the single-bit case: proofs

In the main paper (section 2), we state that, in the single bit case ($b = 1$), the Laplacian, KSH and BRE loss functions over the vector $\mathbf{z}$ of binary codes for each data point can be written in the form of a binary quadratic function without linear term (or a MRF with quadratic potentials only):

$$\min_{\mathbf{z}} E(\mathbf{z}) = \mathbf{z}^T \mathbf{A} \mathbf{z} \quad \text{with} \quad \mathbf{z} \in \{-1, +1\}^N \tag{1}$$

with an appropriate, data-dependent neighborhood symmetric matrix $\mathbf{A}$ of $N \times N$. We can assume w.l.o.g. that $a_{nn} = 0$, i.e., the diagonal elements of $\mathbf{A}$ are zero, since any diagonal values simply add a constant to $E(\mathbf{z})$.

More generally, consider an arbitrary objective function of a binary vector $\mathbf{z} \in \{-1, +1\}^N$ that has the form $E(\mathbf{z}) = \sum_{n,m=1}^N f_{nm}(z_n, z_m)$ and which only depends on Hamming distances between bits $z_n, z_m$. This is the form of the affinity-based loss function used in many binary hashing papers, in the single-bit case. Each term of the function $E(\mathbf{z})$ can be written as $f_{nm}(z_n, z_m) = a_{nm} z_n z_m + b_{nm}$. This fact, already noted by Lin et al. [6], is because a function of 2 binary variables $f(x, y)$ can take 4 different values:

| $x$ | $y$ | $f$ |
|---|---|---|
| 1 | 1 | $a$ |
| $-1$ | 1 | $b$ |
| 1 | $-1$ | $c$ |
| $-1$ | $-1$ | $d$ |

but if $f(x, y)$ only depends on the Hamming distance of $x$ and $y$ then we have $a = d$ and $b = c$. This can be achieved by $f(x, y) = \frac{1}{2}(a - b)xy + \frac{1}{2}(a + b)$, and the constant $\frac{1}{2}(a + b)$ can be ignored when optimizing.

By a similar argument we can prove that an arbitrary function of 3 binary variables that depends only on their Hamming distances can be written as a quadratic function of the 3 variables.

However, this is not true in general. This can be seen by comparing the dimensions of the function spaces spanned by the arbitrary function and the quadratic function. Consider first a general quadratic function $E(\mathbf{z}) = \frac{1}{2}\mathbf{z}^T \mathbf{A} \mathbf{z} + \mathbf{b}^T \mathbf{z} + c$ of $N$ binary variables $\mathbf{z} \in \{-1, +1\}^N$. We can always take $\mathbf{A}$ symmetric (because $\mathbf{z}^T \mathbf{A} \mathbf{z} = \mathbf{z}^T \left(\frac{\mathbf{A} + \mathbf{A}^T}{2}\right) \mathbf{z}$) and absorb its diagonal terms into the constant $c$ (because $z_n^2 = 1$), so we can write

w.l.o.g. $E(\mathbf{z}) = \sum_{n<m}^{N} a_{nm} z_n z_m + \sum_{n=1}^{N} b_n z_n + c$. This has $(n^2 + n + 2)/2$ free parameters. The vector of $2^n$ possible values of $E$ for all possible binary vectors $\mathbf{z}$ is a linear function of these free parameters, Hence, the dimension of the space of all quadratic functions is at most $(n^2 + n + 2)/2$. Consider now an arbitrary function of $b$ binary variables that depends only on their Hamming distances. Although there are $n(n-1)/2$ Hamming distances $d(z_n, z_m)$, they are all determined just by the $n-1$ first distances $d(z_1, z_n)$ for $n > 1$. This is because, given $z_1$, the distance $d(z_1, z_n)$ determines $z_n$ for each $n > 1$ and so the entire vector $\mathbf{z}$ and all the other distances. Also, given the distances $d(z_1, z_n)$ for $n > 1$, the value $z_1 = -1$ produces a vector $\mathbf{z}$ whose bits are reversed from that produced by $z_1 = +1$, so both have the same Hamming distances. Hence, we have $n - 1$ free binary variables (the values of $d(z_1, z_n)$ for $n > 1$), which determine the vector of $2^n$ possible values of $E$ for all possible binary vectors $\mathbf{z}$. Hence, the dimension of the space of all arbitrary functions of Hamming distances is $2^{n-1}$. Since $2^{n-1} > (n^2 + n + 2)/2$ for $n > 5$, the quadratic functions in general cannot represent all arbitrary binary functions of the Hamming distances using the same binary variables.

Finally, note that some objective functions which make sense in the $b$-bit case with $b > 1$ become trivial in the single-bit case. For example, the loss function for Minimal Loss Hashing [7]:

$$L_{\mathrm{MLH}}(\mathbf{z}_n, \mathbf{z}_m;\ y_{nm}) = \begin{cases} \max(\|\mathbf{z}_n - \mathbf{z}_m\| - \rho + 1, 0), & y_{nm} = 1 \\ \lambda \max(\rho - \|\mathbf{z}_n - \mathbf{z}_m\| + 1, 0), & y_{nm} = 0 \end{cases}$$

uses a hinge loss to implement the goal that similar points (having $y_{nm} = 1$) should differ by no more than $\rho - 1$ bits and dissimilar points (having $y_{nm} = 0$) should differ by $\rho + 1$ bits or more, where $\rho \geq 1$, $\lambda > 0$, and $\|\mathbf{z}_n - \mathbf{z}_m\|$ is the Hamming distance between $\mathbf{z}_n$ and $\mathbf{z}_m$. It is easy to see that in the single-bit case the loss $L_{\mathrm{MLH}}(z_n, z_m;\ y_{nm})$ becomes constant, independent of the codes—because using one bit the Hamming distance can be either 0 or 1 only.

# 2 Orthogonality measure: proofs

In paragraph **Are the codes orthogonal?** of the main paper, we define a measure of orthogonality for either the binary codes $\mathbf{Z}_{N \times b}$ or the hash function weight vectors $\mathbf{W}_{b \times D}$, based on the $b \times b$ matrices of normalized dot products, $\mathbf{C}_{\mathbf{Z}} = \frac{1}{N} \mathbf{Z}^T \mathbf{Z}$ and $\mathbf{C}_{\mathbf{W}} = \mathbf{W} \mathbf{W}^T$ (where the rows of $\mathbf{W}$ are normalized), respectively. Here we prove several statements we make in that paragraph.

**Invariance to sign reversals**  Given a matrix $\mathbf{C}$ of $b \times b$ (either $\mathbf{C}_{\mathbf{Z}}$ or $\mathbf{C}_{\mathbf{W}}$) with entries in $[-1, 1]$, define as measure of orthogonality (where $\|\cdot\|_F$ is the Frobenius norm):

$$\perp(\mathbf{C}) = \frac{1}{L(L-1)} \|\mathbf{I} - \mathbf{C}\|_F^2 \in [0, 1]. \tag{2}$$

That is, $\perp(\mathbf{C})$ is the average of the squared off-diagonal elements of $\mathbf{C}$.

**Theorem 2.1.** $\perp(\mathbf{C})$ *is independent of sign reversals of the hash functions.*

*Proof.* Let $\mathbf{S}$ be a $b \times b$ diagonal matrix with diagonal entries $s_{ii} \in \{-1, +1\}$. $\mathbf{S}$ satisfies $\mathbf{S}^T \mathbf{S} = \mathbf{S}^2 = \mathbf{I}$ so it is orthogonal. Hence, $\|\mathbf{I} - \mathbf{S} \mathbf{C} \mathbf{S}\|_F^2 = \|\mathbf{S}(\mathbf{I} - \mathbf{C})\mathbf{S}\|_F^2 = \|\mathbf{I} - \mathbf{C}\|_F^2$. ∎

**Distribution of the dot products of random vectors**  As control hypothesis for the orthogonality of the binary codes or hash function vectors we used the distribution of dot products of random vectors. Here we give their mean and variance explicitly as a function of their dimension.

**Theorem 2.2.** *Let* $\mathbf{x}, \mathbf{y} \in \{-1, +1\}^d$ *be two random binary vectors of independent components, where* $x_1, \ldots, x_d, y_1, \ldots, y_d$ *take the value* $+1$ *with probability* $\frac{1}{2}$. *Let* $z = \frac{1}{d} \mathbf{x}^T \mathbf{y} = \frac{1}{d} \sum_{i=1}^{d} x_i y_i$. *Then* $\mathrm{E}\{z\} = 0$ *and* $\mathrm{var}\{z\} = \frac{1}{d}$.

*Proof.* Let $z_i = x_i y_i \in \{-1, +1\}$. Clearly, $z_i$ takes the value $+1$ with probability $\frac{1}{2}$, so its mean is 0 and its variance is 1, and $z_1, \ldots, z_d$ are iid. Hence, using standard properties of the expectation and variance, we have that $\mathrm{E}\{z\} = \frac{1}{d} \sum_{i=1}^{d} \mathrm{E}\{z_i\} = 0$, and $\mathrm{var}\{z\} = \frac{1}{d^2} \sum_{i=1}^{d} \mathrm{var}\{z_i\} = \frac{1}{d}$. (Furthermore, $\frac{1}{2}(z_i + 1)$ is Bernoulli and $\frac{d}{2}(z + 1)$ is binomial.) ∎

It is also possible to prove that, for random unit vectors of dimension $d$ with real components, their dot product has mean 0 and variance $\frac{1}{d}$.

Hence, as the dimension $d$ increases, the variance decreases, and the distribution of $z$ tends to a delta at 0. This means that random high-dimensional vectors are practically orthogonal. The "random" histograms (black line) in fig. 3 are based on a sample of $b$ random vectors (for $\mathbf{W}$, we sample the component of each weight vector uniformly in $[-1, 1]$ and then normalize the vector). They follow the theoretical distribution well.

# 3    Additional experiments

We show results for an additional hash function and dataset for figures 1 and 2 of the main paper. We also add complete results (using $b = 32$ to 200 bits) for all datasets to the figures 3 and 4 in the main paper, as shown below. The conclusions are as in the main paper:

- Fig. 1: In the paper, we compared ILH diversity mechanisms and their combination with the baseline KSHcut using linear and kernel hash functions, where all the kernel hash functions used the same, common 500 centers for radial basis functions. Here, we compare all the methods using kernel hash functions where each function uses its own 500 centers. Similar to the main paper, diversity methods perform better than (or as good as) the baseline KSHcut and ILHt is clearly the best diversity mechanism, performing much better than KSHcut.

- Fig. 2: We show the results for an additional dataset (CIFAR). Results are similar to the main paper: the highest precision is achieved with a proportion $d/D \approx 50\%$ for ILHf and with $d = D$ for ILHitf, random sampling performs better than bootstrapped sampling for ILHt, and the precision of ILHt continues to increase with the number of bits $b$, greatly exceeding that of KSHcut.

- Fig. 3: the binary codes $\mathbf{Z}_{N \times b} = (\mathbf{z}_1, \dots, \mathbf{z}_b)$ have a wide distribution of dot products. This distribution has some tendency to orthogonality, but it is far from strict orthogonality, or from the distribution of dot products of random vectors, and it seems independent of the number of bits $b$ used.
  As for the hash function weight vectors $\mathbf{W}_{b \times D} = (\mathbf{w}_1, \dots, \mathbf{w}_b)^T$, their distribution of dot products has a stronger tendency to orthogonality, which seems to increase with the number of bits $b$. Hence, although the hash function weight vectors tend to show more orthogonality than the binary code vectors, this is still far from the strict orthogonality demanded by some binary hashing algorithms.

- Fig. 4: ILHt beats all other state-of-the-art methods, or is comparable to the best of them, in different datasets and using different number of bits.

In fig. 5 we also include results for an additional, unsupervised dataset, the Flickr 1 million image dataset [5]. For Flickr, we randomly select $2\,000$ images for test and the rest for training. We use $D = 150$ MPEG-7 edge histogram features. Since no labels are available, we create pseudolabels $y_{nm}$ for $\mathbf{x}_n$ by declaring as similar points its 100 true nearest neighbors (using the Euclidean distance) and as dissimilar points a random subset of 100 points among the remaining points. As ground truth, we use the $K = 10\,000$ nearest neighbors of the query in Euclidean space. All hash functions are trained using $5\,000$ points. Retrieved set: $k$ nearest neighbors of the query point in the Hamming space, for a range of $k$.

The only important difference is that Locality-Sensitive Hashing (LSH) achieves a high precision in the Flickr dataset, considerably higher than that of KSHcut. This is understandable, for the following reasons: 1) Flickr is an unsupervised dataset, and the neighborhood information provided to KSHcut (and ILHt) in the form of affinities is limited to the small subset of positive and negative neighbors $y_{nm}$, while LSH has access to the full feature vector of every image. 2) The dimensionality of the Flickr feature vectors is quite small: $D = 150$. Still, ILHt beats LSH by a significant margin.

In addition to the methods we used in the supervised datasets, we compare ILHt with Spectral Hashing (SH) [8], Iterative Quantization (ITQ) [4], Binary Autoencoder (BA) [2], thresholded PCA (tPCA), and Locality-Sensitive Hashing (LSH) [1]. Again, ILHt beats all other state-of-the-art methods, or is comparable to the best of them, particularly as the number of bits $b$ increases.

# 4 Acknowledgments

Work supported by NSF award IIS–1423515.

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

Figure 1: Extended fig. 1 in the main paper, adding results for a new hash function. Comparing diversity mechanisms and baseline KSHcut using kernel hash functions. Each hash function has its own 500 centers for the radial basis functions. Precision on CIFAR dataset, as a function of the training set size $N$ ($2,\mathbf{000}$ to $20\,000$) and number of bits $b$ (32 to 128).

Figure 2: Extended fig. 2 in the main paper, adding results for a new dataset. All results show precision using a training set of $N = 5\,000$ points of CIFAR dataset. Errorbars over 5 random training sets. Ground truth: all points with the same label as the query. Retrieved set: $k = 500$ nearest neighbors of the query.

Figure 3: Extended fig. 3 in the main paper, adding results for more datasets and for $b = 200$ bits, for both orthogonality of the codes ($\mathbf{C_Z}$ matrix) and of the hash function weight vectors ($\mathbf{C_W}$ matrix). Both matrices $\mathbf{C_Z}$ and $\mathbf{C_W}$ are of $b \times b$ where $b$ is the number of bits (i.e., the number of hash functions).

Figure 4: Extended fig. 4 in the main paper, adding results for $b = 32$ and 200 bits in the comparison with different binary hashing methods in precision and precision/recall, using linear SVMs as hash functions.

Figure 5: Results for the Flickr dataset (unsupervised). The top, middle and bottom panels correspond to figures 2, 3 and 4 in the main paper. Ground truth: the first $K = 10\,000$ nearest neighbors of the query in the original space. Retrieved set: $k = 10\,000$ nearest neighbors of the query.