[Reviews · NeurIPS 2016]

Reviewer 1

Summary

This paper advocates an ensemble learning approach to hashing for retrieval. In particular, instead of optimizing an objective function for hashing that couples all hash bits together (as most hashing papers from the learning community do), this paper learns the functions one at a time, using ideas from ensemble learning. In particular, possible ways to ensure that the learned hash functions are diverse include training each hash function on different subsets of the data, or with different subsets of features. Experiments show the advantages of this approach.

Qualitative Assessment

This is a borderline paper. On the one hand, the authors have done a good job in advocating for an alternative approach to most standard hash learning formulations, via decoupling the hash functions to learn them each independently. The experimental results, which are quite plausible, demonstrate nice properties of this approach, namely that i) hash functions learned in this manner can be quite competitive with existing approaches, ii) orthogonality of hash functions is not really necessary for good performance, iii) we can get substantial training time improvements by using this approach. It's also nice that they've shown that, in the single-bit case, many existing formulations are essentially equivalent. On the other hand, at the end of the day it feels rather incremental. I think the bar is pretty high for new hash learning papers, given the myriad of existing papers that propose similar tweaks on standard formulations. I like the message of the paper, but I'm not sure it's quite to the level of NIPS.

Confidence in this Review

3-Expert (read the paper in detail, know the area, quite certain of my opinion)


Reviewer 2

Summary

The paper describes some methods for better train binary hashing functions by taking advantage of classifier ensembles.

Qualitative Assessment

There are some lacks in the paper. The main usefulness of the paper is not well-explained. The state of the art is not broad enough. Unfortunately, since I don't know much about his subject, I do not know have any other suggestions to provide.

Confidence in this Review

1-Less confident (might not have understood significant parts)


Reviewer 3

Summary

This paper proposes a supervised hashing method by training each hash function independently from each other and introducing diversity among hash functions with techniques from classifier ensembles. Experiments on real datasets are used for evaluation.

Qualitative Assessment

There exist several major issues in this paper. First, the idea to train independent hash functions is not well motivated. It is difficult to know why the proposed method is effective because the hashing problem is totally different from traditional supervised classification problems. Second, the experimental results are unconvincing. The supervised baselines adopted for comparison only contain KSH, which has been outperformed by many recent supervised hashing methods. More advanced supervised hashing methods can be found at [A] [B]. [A] Jun Wang, et al. Learning to Hash for Indexing Big Data - A Survey. Proceedings of the IEEE. 2016. [B] http://cs.nju.edu.cn/lwj/L2H.html

Confidence in this Review

3-Expert (read the paper in detail, know the area, quite certain of my opinion)


Reviewer 4

Summary

This paper presents a simple novel approach to learning supervised binary hash functions for retrieval. This approach considers a series of independent single-bit optimization problems. For each single-bit optimization, a binary bit is assigned to each datapoint to minimize a quadratic binary program, and then a mapping from the datapoint to the corresponding binary bit is learned. The method suggests learning different bits on different datasets to introduce diversity among the bits.

Qualitative Assessment

The main appeal of the paper is its simplicity and its potential applicability to large-scale retrieval applications because of the parallel training algorithm. That said, the training algorithm as presented in the paper (using alternating min-cut [4]) seems slow for tens of thousands of data-points. Alternative optimization for solving the single-bit assignment problem can be used. The main drawback of this approach to me is that solving the binary bit assignment problem independently from the family of hash functions considered, may result in a bit assignment that is very difficult to learn a hash function for. This issue should be discussed in the paper. Pluses: + Simplicity + Large scale applicability + Reasonable experimental results Negatives: - Solving the bit assignment problem independently from the family of hash functions may result in a bit assigment that is very difficult to learn a hash function for. - The diversity promoting setup seems hacky. More direct way of promoting diversity is desired In summary, given the simplicity of the proposed approach and it's competitive experimental results, I think publishing this paper at NIPS will benefit large scale retrieval applications. More comments: Consider citing and potentially comparing with these papers: * Fast pose estimation with parameter sensitive hashing, ICCV, 2003. Uses adaboost to sequentially learn binary bits * Hamming distance metric learning, NIPS, 2012. Uses a continuous upper bound on the discrete objective for ease of optimization

Confidence in this Review

3-Expert (read the paper in detail, know the area, quite certain of my opinion)


Reviewer 5

Summary

The paper proposes a novel supervised binary hashing method. Instead of optimizing over all bits of a binary code as the final objective, the proposed method learns hash functions independently for each bit. To avoid a trivial solution it borrows ideas from ensemble learning and introduces a notion of diversity (instead of orthogonality common in classical hashing literature) among bits by a bootstrapping type mechanism. The final algorithm can be optimized more efficiently since it is more amenable to parallelization and outperforms similar supervised hashing algorithm such as KSH.

Qualitative Assessment

The proposed scheme seems to be novel in hashing literature despite similarities to ensemble learning. It is both interesting and surprising that training hash functions independently can achieve better retrieval performance. While the heuristics seem to be fine, a more rigorous theoretical justification of why optimizing each bit independently leads to better retrieval results would be very interesting.

Confidence in this Review

2-Confident (read it all; understood it all reasonably well)


Reviewer 6

Summary

Binary hashing can improve information retrieval efficiency from both time and space perspectives. Most existing methods learn hashing codes and function by a composite joint optimization, which makes the optimization a hard problem. Inspired by the diversity requirement in classifier ensemble, this paper proposes to learn hash function for each hashing code independently. Three types of diversity strategies are used, namely on different initializations, different training sets and different feature subsets. The new approach benefits from its computational efficiency and nesting property. Diverse experiments validate the effectiveness of the proposed ILH strategy.

Qualitative Assessment

Inspired by diversity requirement to train a good classifier in an ensemble scenario, this paper proposes to train binary hashing codes independently. This Independent Laplacian Hashing (ILH) approach optimizes in a two-step way: first learns hash codes and then trains the hashing function. Three types of diversity are proposed, and from the experiment results the one learned with different training sets can achieve better performance. This paper investigates ILH approach from different perspectives, and explains the results in detail. Some discussions about orthogonal codes are interesting. Here are some suggestions/questions about the experiments: 1. Statement in line 186-188 is not very clear. 2. In most experiments, only precision is reported. For example, in Fig.1, its better to show the change of recall at the same time. 3. In Fig.1, different methods are plotted in different graphs. Ploting different approaches with the same b value in a single graph may be better to find their differences. 4. Since ILH method is based on the random initialization, training subset and feature subset, different trials will get various results. How to control experiments w.r.t. this randomness is important. Will different random trails all output good last results or the results are based on a specific randomize strategy? 5. Performance of ILHf relies on the property of features. It discards useful features when training on CIFAR dataset, will it perform better when there are irrelevant or redundant features? 6. Time comparison on different datasets (about different versions of ILH and other compared methods) may be necessary to show the property of the proposed method.

Confidence in this Review

2-Confident (read it all; understood it all reasonably well)